# Inhibition of Chitosan with Different Molecular Weights on Barley-Borne *Fusarium*
*graminearum* during Barley Malting Process for Improving Malt Quality

**DOI:** 10.3390/foods11193058

**Published:** 2022-10-01

**Authors:** Jing Luan, Xu Wei, Zhefeng Li, Wenzhu Tang, Fan Yang, Zhimin Yu, Xianzhen Li

**Affiliations:** School of Biological Engineering, Dalian Polytechnic University, Ganjingziqu, Dalian 116034, China

**Keywords:** antifungal, barley, chitosan, *Fusarium graminearum*, malting

## Abstract

There are many *Fusarium graminearum* contaminations in barley that are often associated with malt and beer quality issues. Thus, it is important to find a biological antifungal agent to prevent the growth of *F. graminearum* during malting. Minimum inhibition concentration (MIC) of chitosan for mycelial growth and spore germination of *F. graminearum* was 2.6 g/L and 1.6 g/L, respectively, indicating that the *F. graminearum* strain was highly sensitive toward chitosan. Chitosan with a molecular weight of 102.7 kDa was added at 0.5 g/kg during the first steeping stage, resulting in the maximum inhibition rate of *F. graminearm* in barley. The biomass of *F. graminearm* and deoxynivalenol content in the infected barley at the end of germination with 0.5 g/kg chitosan treatment were decreased by 50.7% and 70.5%, respectively, when compared with the infected barley without chitosan. Chitosan could remove the negative effects of *F. graminearm* infection on barley germination and malt quality, which makes the application of chitosan during the steeping process as a potential antifungal agent in the malting process to protect from *F. graminearum* infection.

## 1. Introduction

Most beer in the world is brewed from barley malt [1]. The preparation of malt, i.e., malting, is a natural physiological and biochemical process of barley seed germination [2]. Barley germination requires a variety of conditions, including sufficient oxygen, suitable temperature, high humidity and rich grain nutrients, which are also favorable conditions for the growth and multiplication of microorganisms in barley [3]. Microorganisms in barley can easily interact with grain metabolism during the malting process [4]. Therefore, microorganisms in barley have an important impact on the malting process and the final malt quality [5].

A filamentous fungal community in barley seeds is formed before harvest, during storage and during the malting process [6]. Higher total rainfall during the planting process or grain harvest can lead to a sharp increase in fungal growth (especially the *Fusarium* genus), which may be related to quality defectsinmalt [7]. *Fusarium graminearum* is an important member of the barley fungal community and causes problems related tobarley germination and malt quality [8]. High incidence of *F. graminearum* contamination has been widely found in various crops worldwide [9]. *F. graminearum* can directly cause premature yeast flocculation, beer gushing, wort filtration problems and visible beer hazes, which are major defects of the beer brewing process and beer quality [10]. Furthermore, *F. graminearum* can produce mycotoxins, such as deoxynivalenol (DON), which are producedduring fungal growth along with malting; they possess carcinogenic and mutagenic potential and can be accumulated into malt and beer products [8,11].

Inhibiting the infection and reproduction of the *Fusarium* strain during malting is an effective measure in improving the quality of malt and the accumulation of mycotoxins in beer [12]. At present, the methods used to inhibit the growth and reproduction of molds in barley mainly include spraying chemical fungicides [1], physical treatment [11] and adding antifungal microorganisms [2,5]. However, excessive use of chemical fungicides can cause food safety issues and endanger workers’ health. It is difficult to treat barley by radiation methods on an industrial scale. Adding microorganisms in the malting process not only increases the production cost, but also may cause secondary pollution. Therefore, the development of environmentally friendly, efficient and inexpensive antifungal agents is a key problem that needs to be solved urgently.

Chitosan is a natural polycationic biopolymer derived from the deacetylation of chitin, which can be obtained from crustacean shells or certain fungi by chemical or microbial catalysis [13,14]. Thus, chitosan, as a potential important resource, has attracted great attention in food processing due to its biological properties such as nontoxicity, biocompatibility and biodegradability [15,16]. Both in vitro and in situ studies have shown that chitosan had bactericidal effects on many plant-pathogenic fungi [13,17]. Chitosan also has inhibitory effects on soil-borne pathogenic fungi including *Fusarium* pathogens [18,19].

According to the available information, there is no investigation on the use of chitosan in malting for the inhibition of barley-pathogenic fungi and improvement of malt quality. Therefore, the main aim of this study was to evaluate and optimize the inhibitory effect of chitosan on *F. graminearum* infection in barley malt. The secondary aim was to examine the optimized chitosan treatment efficacy on decontamination of malt contaminated with *F. graminearum* and DON. The third aim was to evaluate the effect of chitosan on barley germination and malt quality.

## 2. Materials and Methods

### 2.1. Microorganisms and Materials

*Fusarium graminearum* BF-08, originally isolated from brewing barley, was obtained from the collection in our laboratory. *F. graminearum* was incubated on potato dextrose agar (PDA) and chilled at −80 °C.

Winter 2-row barley (*Hordeum vulgare* L.) was harvested from Canada in October 2016, and was kindly provided by Hailaer Agricultural Reclamation Co., Ltd. (Hulunbeier, China).

Four chitosan samples with different molecular weights (Mw) (5.1 kDa, 30.2 kDa, 102.7 kDa and 189.3 kDa) were purchased from Zhejiang Golden-Shell Biochemistry Co., Ltd. (Wunzhou, China). The deacetylation degree of chitosan is 90%. An amount of 40 g of chitosan was dissolved in 1 L of acetic acid solution (1%, *v*/*v*) by stirring overnight at room temperature. The original chitosan solution (40 g/L) was then diluted to different concentration solutions, and the pH value of each chitosan solution was adjusted to 5.5 with 1 mol/L NaOH.

### 2.2. Determination of Minimum Inhibition Concentration (MIC)

The inhibitory effect of chitosan on *F. graminearum* growth was carried out according to theagar dilution method as described by Al-Hetar et al. [20] and De Rodríguez et al. [21]. Chitosan solutions were sterilized using a sterile ultrafiltration membrane (0.22 μm, Millipore Millex-GS, Merk, Darmstadt, Germany). The sterile chitosan solutions and PDA were subsequently combined to obtain concentrations of 0.5, 1.0, 2.0, 5.0, 10.0 and 20.0 g/L, and were poured into sterile Petri dishes (9 cm in diameter). Control plates consisted of PDA only. Afterwards, a 7 mm diameter disk cut from the periphery of a 7-day-old culture of *F. graminearum* strain was placed in the plates’ centers. The plates were incubated at 30 °C for 3 days. The diameter of mycelial growth, taking two values in cross for each plate, was measured with vernier calipers after being cultivated for 24 h, 48 h and 72 h. The inhibition rate of mycelial growth was determined with reference to the control concentration growth, and calculated according to the formula:$$\mathrm{Inhibition}\;\mathrm{rate}\;\left(\%\right)=\left(\frac{\mathrm{DC}-\mathrm{DT}}{\mathrm{DC}}\right)\times 100$$
where DT is the colony diameter of different chitosan concentrations, and DC is the colony diameter of the control concentration.

Spore germination was determined by spreading 300 μL spore suspension (300 spore/mL) onto PDA plates and adding different concentrations of chitosan (0, 0.5, 1.0, 2.0, 5.0, 10.0 and 20.0 g/L). The plates were incubated at 30 °C and the number of colonies was counted after being cultivated for 48 h. The inhibition rate of spore germination was determined with reference to the control concentration growth, and calculated according to the formula:$$\mathrm{Inhibition}\;\mathrm{rate}\;\left(\%\right)=\left(\frac{\mathrm{NC}-\mathrm{NT}}{\mathrm{NC}}\right)\times 100$$
where NT is the colony number of different chitosan concentrations, and NC is the colony number of the control concentration.

MIC values (concentration causing 90% inhibition of mycelial growth or spore germination) were calculated by using the probit analysis method as described by Im et al. [22]. Statistical evaluation was performed using the SAS Probit Analysis Program (Version 9.4, SAS Institute Inc., Raleigh, NC, USA). MIC values were calculated statistically and were significant associated with chi-square values at a *p* < 0.05 level.

### 2.3. Barley Grain Surface Disinfection

Barley grains were sanitized by following the method as described by Oliveira et al. [2] with some modifications. The barley grains were disinfected by using 10% (*w*/*v*) H_2_O_2_ solution for 10 min with continuous stirring. Subsequently, the barley grains were washed for 5 min twice with 500 mL of sterile water.

### 2.4. Mold Spore Suspension Preparation and Grain Inoculation

The spore suspension of *F. graminearum* BF-08 was prepared by following the method as described by Mauch et al. [23]. *F. graminearum* BF-08 was grown on PDA plates at 25 °C for 5 days. A small piece of *F. graminearum* PDA was inoculated in synthetic nutrient-poor bouillon (SNB), and then it was grown at 25 °C for 14 days with 120 rpm stirring to induce spore production. The concentration of *F. graminearum* spores was determined with a hemocytometer, and was diluted to approximately 10^5^ spore/mL with sterile water.

Infected barley grains (IBG) were prepared by following procedure as described by Oliveira et al. [8]. Barley grains were inoculated with 2% (*v*/*w*) of *F. graminearum* macroconidia suspension, and were homogeneously mixed. The inoculated grains were cultivated in a temperature- and humidity-controlled incubator (Zhicheng, Shanghai, China) for 5 days at 25 °C with 98% relative humidity to enable mold proliferation on the grains. During the period of culture, the grains were mixed evenly every 12 h. IBG were obtained after 5 days of culture and prepared for malting. Uninfected barley grains (UBG) were processed in the same way without mold inoculation as a control.

### 2.5. Barley Malting Process

The mixed barley grains (MBG) were obtained by mixing 40 g IBG with 160 g UBG, which were malted to produce mixed malt grains (MMG). UBG were malted to obtain the uninfected malt grains (UMG) as a control. The barley grains (200 g) were malted by steeping, germination and kilning in a micro-malting system (Buhler, Germany). The barley grains were steeped for 32 h (steeping in water for 6 h, and then resting in air for 18 h, steeping in water for 4 h, and resting in air for 4 h) and germinated for 96 h. The air-rest and germination stages were carried out at 16 °C and 98% relative humidity. The kilning of green malt grains was carried out for 8 h at 40 °C, then 8 h at 60 °C, followed by 8 h at 80 °C. The rootlets of kilned malt were removed mechanically and then used for malt quality analysis.

### 2.6. Influence of Chitosan on Fusarium graminearum Infection during Malting

To evaluate whether chitosan could affect *F. graminearum* infection during the malting process, the MBG were germinated as described above after chitosan with a molecular weight of 5.1, 30.2, 102.7 or 189.3 kDa (0.5 g/kg barley grains) was added to the steeping water in the first steeping stage. To examine the effect of the concentration of chitosan on *F. graminearum* infection during barley germination, 0.05–1 g chitosan (102.7 kDa)/kg MBG was added to the water used in the first steeping stage. To investigate the influence of the method of steeping on *F. graminearum* infection, 102.7 kDa chitosan (0.5 g/kg barley grains) was added to the steeping water either in the first steeping stage, the second steeping stage or both the first and the second steeping stages, then the MBG were germinated as above.

### 2.7. Analysis of the Infection Rate of Fusarium graminearum

One hundred MBG kernels, after 96 h germination, were aseptically put into 20 standard Petri dishes (5 kernels/dish) containing half-strength PDA and were incubated at 30 °C for 5 days [24]. The molds on barley kernels were identified to the genus level and counted after 5 days of culture.

### 2.8. Quantification of Fusarium graminearum Growth by PCR and Photometric Assay

PCR and photometric assays were carried out to determine the biomass of *F. graminearum* on MBG during malting by following the method as described by Oliveira et al. [25] with some modifications. Total genomic DNA of *F. graminearum* was extracted as follows: 20 g of barley grains were added into 100 mL of lysis buffer [2.5 mL/L Triton-X-100, 2.5 mmol/L trisaminomethane (Tris), 125 mmol/L guanidine-HCl, 50 mmol/L NaCl, 5 mmol/L ethylenediaminetetraacetic acid (EDTA)] in sterilized Erlenmeyer flasks, stirred at 120 rpm for 15 min and rested for another 15 min. The supernatant (1.2 mL) was centrifuged for 10 min at 8000× *g* and the resulting precipitates were collected. *F. graminearum* DNA was isolated by using a High Pure PCR Template Preparation Kit (Roche, Shanghai, China).

The PCR reaction conditions were as follows: The genomic DNA (2 μL) was added into a reaction mixture (48 μL) containing 1× reaction buffer, 2.5 mmol/L deoxyribonucleotide triphosphate (d NTP), 2.5 units *Taq* Polymerase and 25 pmol forward and reverse primer. Species-specific primers of *F. graminearum* gene [16F, (forward primer: 5′-CTCCGGATATGTTGCGCAA-3′) and 16R, (reverse primer: 5′-GGTAGGTATCCGACATGGCAA-3′)] were used (TaKaRa Bio, Dalian, China). The amplification of DNA was conductedin a 96-well block (TProfessional Basic Thermocycler (Roche Light Cycler 480, Basel, Switzerland)) under the condition of an initial 5 min denaturation cycle at 95 °C, followed by 40 cycles of 20 s at 95 °C, 20 s at 65 °C, and 30 s at 72 °C, with a final extension cycle of 5 min at 72 °C.

A Qubit^®^ 1.0 Fluorometer (Invitrogen) was used to quantify PCR products by using a Qubit dsDNA BR Assay kit (Q32853 Invitrogen). The samples were processed according to the manufacturer’s instructions, and then were recorded after incubation for 2 min at 22 °C. The concentration of samples was calculated in μg/mL (elution buffer). The samples were diluted to fit a linear range (10–30 μg/mL).

### 2.9. Analysis of Deoxynivalenol (DON) Content

DON in barley was exacted and measured according to the method described by Boenisch and Schäfer [26] with some modifications. An amount of 10 g of barley samples were pestled in liquid nitrogen and homogenates were diluted with 50 mL of sterile water. Samples were shaken for 30 min at 150 rpm and 25 °C for DON extraction in barley. The barley debris wa scentrifuged for 10 min with 4000× *g* and 25 °C, and the resulting supernatant was collected. The concentration of DON was detected by using the DON rapid ELISA kit (ZZBio, Shanghai, China). The extinction was measured by photometry at 450 nm by an ELISA reader (Molecular Devices, Shanghai, China). The concentration of DON was analyzed with a standard curve of DON concentration.

### 2.10. Analysis of Barley Germination Performances and Malt Quality Parameters

To assay the barley germination rate, 100 kernels were randomly sampled from themicro-malting system during barley germination. When the first visible sign of the root appeared, the barley grains were scored visually as germinated [27]. Barley germination trials were operated in triplicate for these determinations.

Water content and brittleness of malt were determined according tothe analysis method of ASBC [28]. Wort was produced according tothe standard European Brewery Convention (EBC) congress mash (method 4.5.1) [29]. The extract, free α-amino nitrogen, β-glucan, diastatic power, Kolbachindex, viscosity, turbidity, color intensity, glucose content, maltose content, maltritose content, total acid and pH of wort were determined according tothe analysis methods of ASBC [28].

### 2.11. Statistical Analysis

All tests were carried out in triplicate. The test values were expressed as the mean ± standard deviation. Analysis of variance and significant differences among means were tested by a dependent-sample *t* test (*p* < 0.05) using SPSS software (version 17.0, SPSS Inc., Chicago, IL, USA).

## 3. Results and Discussion

### 3.1. Inhibitory Effects of Chitosan against Fusarium graminearum

The *F. graminearum* strain is known to be a dominant contaminating strain in brewing barley, and can cause serious issues affecting barley germination and malt quality [30]. The minimum inhibition concentration (MIC) of chitosan against the mycelial growth and spore germination of *F. graminearum* BF-08 under in vitro conditions was determined and is shown in Table 1. The results indicate that the *F. graminearum* strain was highly sensitive toward chitosan because the MIC of chitosan against the mycelial growth of the *F. graminearum* strain was between 2.6 and 12.8 g/L, and the MIC of chitosan against the spore germination of the *F. graminearum* strain was between 1.6 and 3.6 g/L. Similar results on the antifungal activity of chitosan against pathogens and spoilage molds were reported by Al-Hetar et al. [20] for *F. oxysporum* f. sp. *cubense*, a causal agent of banana wilt; by Dananjaya et al. [31] for *F. oxysporum,* a pathogenic fungusisolated from zebrafish; by Kheiri et al. [32] for *F. graminearum*, a wheat seed-bornefungus; and by Khan and Doohan [33] for *F. culmorum*, causing Fusarium head blight disease of wheat and barley.

Chitosan can inhibit the cell growth of some bacteria, fungi and yeast. This antimicrobial activity depends on the molecular weight (Mw) of chitosan [34]. As shown in Table 1, the MIC of chitosan against the *F. graminearum* strain was significantly decreased (*p* < 0.01) with the increase in the Mw of chitosan, which was probably due to the fact that the number of primary amino groups, which can make stronger interactions with mold cells, present in chitosan tends to increase upon the increase in the Mw of chitosan [35]. There is a general tendency observe an increase in the antifungal activity of chitosan with an increase in its Mw [36]. Zhang and Zhu [37] reported that the antifungal effect of chitosan with Mw below 300 kDa was enhanced upon increasing Mw. Hernández-Lauzardo et al. [38] indicated that the inhibition of chitosan on shape, sporulation and germination of mold spore increased with an increase in Mw. Younes et al. [39] investigated the effect of chitosan with different Mw (43–135 kDa) on *F. oxysporum*. They reported that the antifungal activity increased with an increase in Mw, particularly when Mw was larger than 100 kDa.

Chitosan can effectively inhibit the development of the *F. graminearum* strain at different life-cycle stages. The spore germination of the *F. graminearum* strain tested was more sensitive to chitosan than mycelial growth (Table 1). Similar results were also observed by Liu et al. [40]. They found that the spore germination of *Penicillium expansum* was completely inhibited by chitosan at 0.5%, while the mycelial growth of *P. expansum* was not completely inhibited by chitosan at 1.0%. Al-Hetar et al. [20] also found that the MIC of chitosan against the mycelial growth and sporulation of *F. oxysporum* f. sp. *cubense* was 26 g/L and 8 g/L, respectively. Therefore, chitosan should be used at the beginning stage of barley germination because, at this time, the spores of the *F. graminearum* strain in barley begin to germinate from dormancy by barley steeping.

### 3.2. Effects of Chitosan on Infection Rate of Fusarium graminearum in Barley

A spore suspension of *F. graminearum* BF-08 was inoculated into barley grains and then cultured at 25 °C and 98% relative humidity. The infection mode was observed visually for 5 days for mold growth and covering of the barley surface. We obtained a specific initial degree of infection on barley grains (20%), which simulated the field contamination and the spread of *F. graminearum* infection during malting.

For mixed barley grains (MBG), the infection rate of *F. graminearum* increased from 21% to 58% at the end of steeping. Inoculation of barley resulted in a 100% infection rate, and there was essentially no change through the rest of the germination. Increased temperature during kilning reduced the infection rate from 100% to 40% (Figure 1A–C).

The infection rate of *F. graminearum* in barley was monitored when chitosan with different Mw was added to the steeping water during the malting process. As shown in Figure 1A, MBG treated by chitosan with Mw of 5.1 kDa, 30.2 kDa, 102.7 kDa and 189.2 kDa during the steeping process exhibited significantly lower (*p* < 0.05) infection rates throughout the malting process. The infection rate of *F. graminearum* in barley decreased with an increasing in the Mw of chitosan. The best inhibition effect was obtained when chitosan with Mw of 102.7 kDa and 189.2 kDa was added to the steeping water, and the infection rate was reduced to 53.2% and 54.0%, respectively, at the end of barley germination (124 h of malting). The inhibition effect of *F. graminearum* infection during malting was the best when Mw of chitosan was larger than 100 kDa.

The infection rate of *F. graminearum* BF-08 in barley was monitored when different concentrations of chitosan were added to the steeping water during the malting process. In the present study, the infection rate throughout the malting process by 0.05–1 g chitosan/kg barley grain treatment was markedly lower (*p* < 0.05) than the infection rate of MBG (Figure 1B). The infection rate of *F. graminearum* in barley decreased with an increase inthe concentration of chitosan. As for the optimum inhibition effect, the infection rate reduced to 54.4% and 53.2%, respectively, at the end of barley germination (124 h of malting); this was obtained when 0.5 g/kg and 1 g/kg chitosan were added to the steeping water. The inhibition effect of *F. graminearum* infection during malting was the best when amount of chitosan added to the steeping water was greaterthan 0.5 g per kg of barley grains.

The infection rate of *F. graminearum* BF-08 in barley was monitored when chitosan was added in different steeping processes. As shown in Figure 1C, MBG treated with chitosan in the first steeping stage, the second steeping stage or in both steeping stages exhibited significantly lower (*p* < 0.05) infection rates throughout the malting process. The optimum inhibition effect was obtained when chitosan was added in the first steeping stage or both steeping stages. The infection rate reduced to 51.2% at the end of barley germination (124 h of malting) when chitosan was added in the first steeping stage.

Taken together, the antifungal results indicate that chitosan could potentially be applied in the barley malting industry to prevent malt from infection catalyzed by the *F. graminearum* strain. Chitosan with a Mw of 102.7 kDa, added at 0.5 g/kg during the first steeping stage, led to the maximum inhibition of *F. graminearum* infection.

### 3.3. Effects of Chitosan on Fusarium graminearum Biomass and Deoxynivalenol (DON) Content in Barley

The barley grains used for malting may contain a certain amount of *F. graminearum* contamination. Therefore, an increase in the water content of barley during the malting process not only promotes the seed germination, but also promotes microorganism growth [8]. In this study, a mix of infected barley grains (IBG) (20%) with uninfected barley grains (UBG) (80%) was used to evaluate the spread of *F. graminearum* infection, fungal growth and DON production during malting. As shown in Figure 2, the *F. graminearum* biomass of MBG during malting with chitosan treatment was significantly lower (*p* < 0.05) than the biomass of MBG. The maximum *F. graminearum* biomass at the end of germination with 0.1 g/kg and 0.5 g/kg chitosan treatment was 42.8% and 50.7% lower, respectively, than the corresponding results of MBG. This result indicates that chitosan could effectively control the propagation of *F. graminearum* during malting.

The interest to brewers and malt producers in the occurrence of *F. graminearum* infection is the mycotoxin DON. DON is one of the primary trichothecene toxins produced by *F. graminearum* during barley storage and germination [41]. In general, DON is associated with vomiting, feed refusal and also malt quality defects and beer gushing [7]. Figure 3 showsthat effects of chitosan on DON content in MBG during malting. Compared with MBG, DON content during malting with chitosan treatment decreased dramatically (*p* < 0.05). Furthermore, the maximum DON content, found at the end of germination, with 0.1 g/kg and 0.5 g/kg chitosan treatment, was 64.1% and 70.4% lower, respectively, than the corresponding results of MBG. Khan et al. [33] also reported that chitosan was effective in reducing DON contamination of wheat and barley caused by *F. culmorum*. This decrease in DON levels may be associated with a decrease in *Fusarium* biomass and the inhibition of mycotoxin biosynthesis by chitosan treatment.

### 3.4. Effects of Chitosan on Germination Ability of Fusarium graminearum-Infected Barley

The barley germination rate of MBG was determined in the malting process when 0.1 g/kg and 0.5 g/kg chitosan (Mw of 102.7 kDa) was present in the steeping water for the first steeping stage (Figure 4). Compared with UBG, the germination rate of MBG throughout the malting process was markedly reduced (*p* < 0.05). The barley germination rate of MBG throughout the malting process had a notable improvement (*p* < 0.05) when treating with chitosan in the first steeping stage. The final barley germination rate of MBG with 0.1 g/kg and 0.5 g/kg chitosan treatment was 5% and 4% lower, respectively, than that of UBG, which indicated that only a small amount mold on barley could still inhibit barley germination. This finding correlates with previous studies of *Fusarium* strains in which barley inoculated with *F. graminearum* during malting had a reduction in germinative energy [12]. Loss of germination ability was negatively correlated with an improvement in grain water sensitivity caused by mold infection (data not shown). Furthermore, the abundant DON was synthesized and secreted by an infected *F. graminearum* strain that could inhibit barley germination [7]. Thus, chitosan could be used to improve the germination ability by suppressing the infection and reproduction of an infected *F. graminearum* strain in barley. On the other hand, chitosan could also be used as an elicitor in seed priming to enhance the germination ability of barley [27,42].

### 3.5. Effects of Chitosan on Quality Performance of Fusarium graminearum-Infected Malt

The purpose of malting is to stimulate the activity of hydrolase, promote the modification of endosperm, provide sugar extract and amino acids for beer brewing, and provide color and flavor for beer. High malt quality is dependent not only on the absence of any mycotoxin, but also on the parameters of the malting and malt quality. Therefore, standard procedures are usually used to evaluate the properties of malt and wort [29].

MBG was malted to produce mixed malt grains (MMG). The characteristic visual aspects of malt grains from MBG and MBG treated with chitosan are shown in Figure 5. MMG contained large amounts of malt grains with a red-pink color, which is typically associated with *F. graminearum* infection. However, MMG treated with chitosan had a golden yellow color, which is typically associated with healthy cereal tissue.

The impact of chitosan on the quality parameters of MMG was determined in Table 2. The infection with *F. graminearum* brought upon many negative impacts to the malt quality parameters. *F. graminearum*-infected barley led to some malt quality parameters, including brittleness, Kolbachindex, free α-amino nitrogen, extract, diastatic power, glucose content, maltose content and maltotriose content, being reduced significantly (*p* < 0.05), and others, including colority, turbidity and β-glucan content, were increased significantly (*p* < 0.05). The malt quality parameters of MMG had a notable improvement (*p* < 0.05) by treating with chitosan in the first steeping stage. The brittleness of MMG treated with 0.1 g/kg and 0.5 g/kg of chitosan had 3.8% and 4.8%, respectively, lower values than those of UMG. However, MMG treated with 0.1 g/kg and 0.5 g/kg of chitosan had β-glucan content that was significantly lower (*p* < 0.05) and free α-amino nitrogen and maltose contents that were significantly higher (*p* < 0.05) when compared with UMG, indicating that chitosan could promote barley modification and eliminate the negative effects of molds on malt quality. These results suggest that the product qualities can be improved when chitosan is used as antifungal agent during malting. The advantages of using chitosan are obvious; chitosan can be produced in a sustainable way, it is easy applicable, is highly specific against *Fusarium* infection and improves the quality parameters of the final product.

## 4. Conclusions

This study evaluated *Fusarium* infection, DON content, barley germination and malt quality in chitosan-treated barley. The results found that the *F. graminearum* strain was highly sensitive towards chitosan. The infection rate, biomass and DON content of *F. graminearum* MBG during malting tread with chitosan was significantly decreased (*p* < 0.05) when compared to values of MBG. Furthermore, chitosan could remove the negative effects of *F. graminearum* infection on germination and malt quality parameters. Therefore, chitosan could be potentially applied in the malting industry as an edible antifungal agent by adding it tothe steeping process to avoid malt quality issues caused by *F. graminearum* infection. We found that chitosan can promote barley germination and improve malt quality, but its optimal technology conditions arecurrently unknown, and we will continue to study thesein the future. At the same time, we will further expand the experimental scale to verify the application effect of chitosan, so as to better apply chitosan to the malting industry.

## Figures and Tables

**Figure 1 foods-11-03058-f001:**
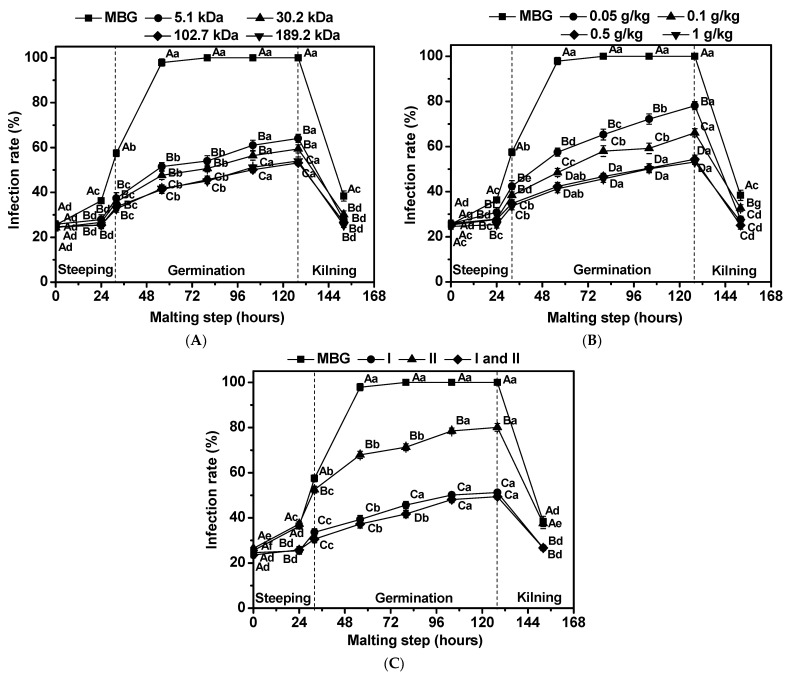
Effects of chitosan with different Mw (**A**), different concentrations (**B**), and different treatments (**C**) on *Fusarium*
*graminearum* infection rate in barley during malting. Each number represents the mean ± standard deviation of three replicates. Within the same malting time and among different treatment conditions, different capital letter superscripts indicate significant differences (*p* < 0.05); within the same treatment condition and among different malting times, different lowercase superscripts indicate significant differences (*p* < 0.05).

**Figure 2 foods-11-03058-f002:**
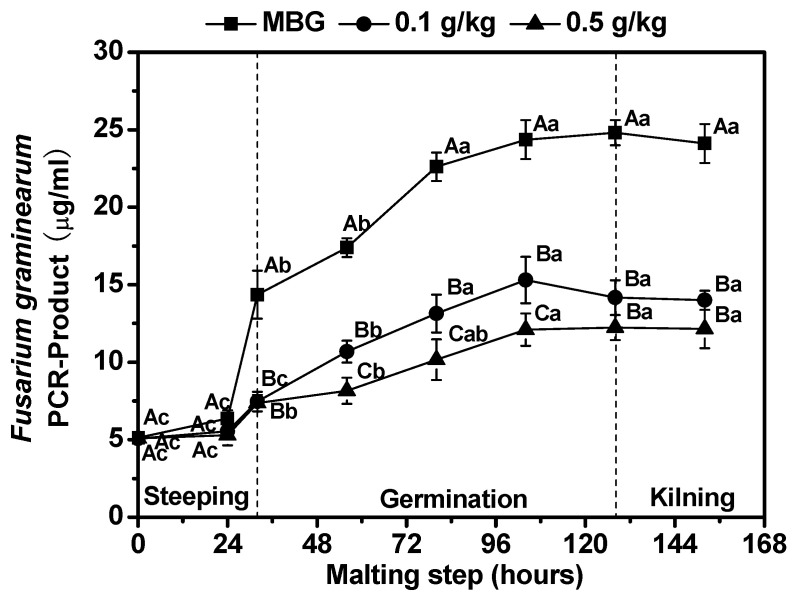
Effects of chitosan on *Fusarium graminearum* biomass in barley during malting. Each number represents the mean ± standard deviation of three replicates. Within the same malting time and among different treatment conditions, different capital letter superscripts indicate significant differences (*p* < 0.05); within the same treatment condition and among different malting times, different lowercase superscripts indicate significant differences (*p* < 0.05).

**Figure 3 foods-11-03058-f003:**
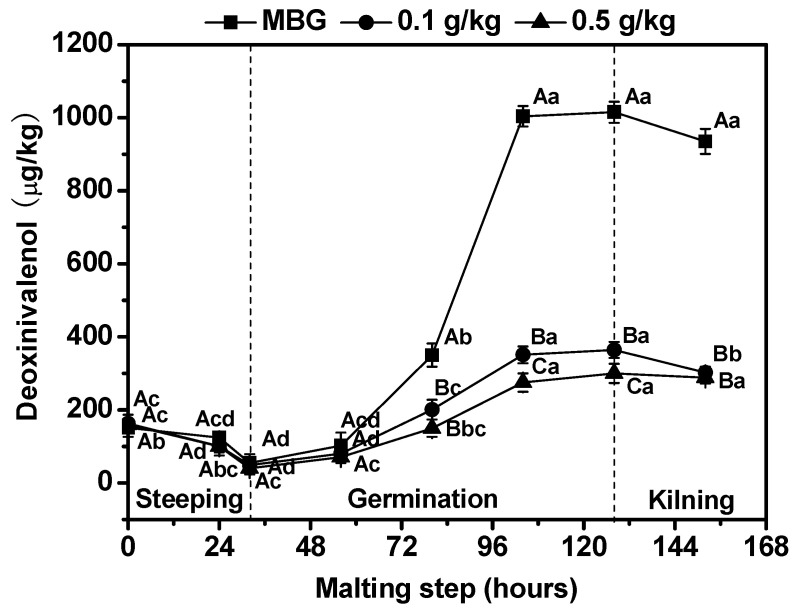
Effects of chitosan on deoxynivalenol (DON) content on barley during malting. Each number represents the mean ± standard deviation of three replicates. Within the same malting time and among different treatment conditions, different capital letter superscripts indicate significant differences (*p* < 0.05); within the same treatment condition and among different malting times, different lowercase superscripts indicate significant differences (*p* < 0.05).

**Figure 4 foods-11-03058-f004:**
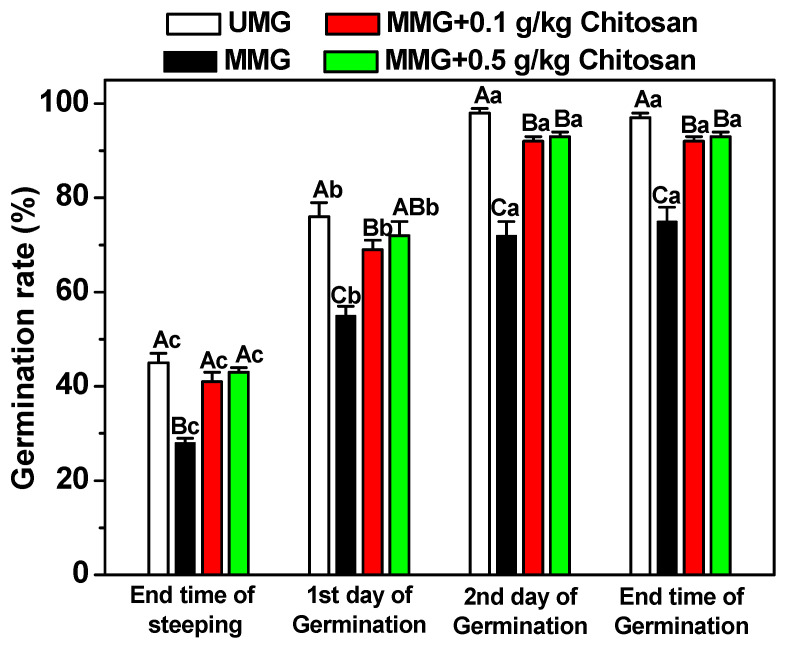
Germination rate of uninfected malt grains (UMG), mixed malt grains (MMG) and MMG treated with 0.1 g/kg of chitosan or 0.5 g/L of chitosan. Each number represents the mean ± standard deviation of three replicates. Within the same malting step and among different treatment conditions, different capital letter superscripts indicate significant differences (*p* < 0.05); within the same treatment condition and among different malting steps, different lowercase superscripts indicate significant differences (*p* < 0.05).

**Figure 5 foods-11-03058-f005:**
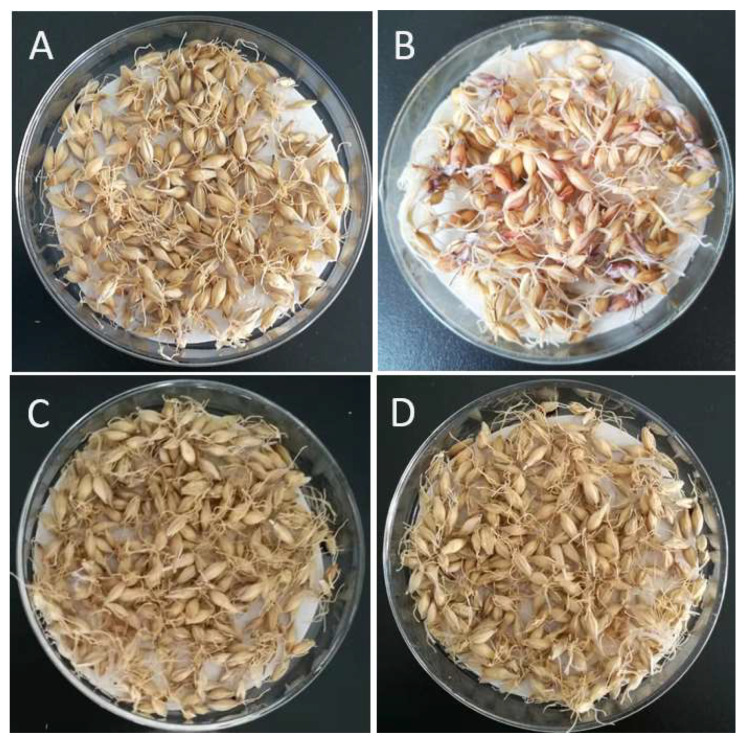
Visual aspects of uninfected malt grains (UMG) (**A**), mixed malt grains (MMG) (**B**) and MMG treated with 0.1 g/kg of chitosan (**C**) or 0.5 g/kg of chitosan (**D**).

**Table 1 foods-11-03058-t001:** The minimum inhibition concentration (MIC) (g/L) of chitosan with different molecular weights (Mw) against the mycelial growth and spore germination of *Fusarium*
*graminearum* isolated from barley.

Chitosan (k Da)	Mycelial Growth	Spore Germination
5.1	12.8 ± 0.4 ^a^	3.6 ± 0.2 ^c^
30.2	4.6 ± 0.1 ^b^	2.0 ± 0.1 ^f^
102.7	3.1 ± 0.2 ^d^	1.6 ± 0.1 ^g^
189.3	2.6 ± 0.1 ^e^	1.6 ± 0.1 ^g^

Each number represents the mean ± standard deviation of three replicates. Mean values with different superscript letters indicate a significant difference (*p* < 0.05).

**Table 2 foods-11-03058-t002:** The quality parameters of uninfected malt grains (UMG), mixed malt grains (MMG) and MMG treated with 0.1 g/kg or 0.5 g/kg of chitosan (102.7 kDa) in the first steeping stage.

Parameters	UMG	MMG	MMG + 0.1 g/kg Chitosan	MMG + 0.5 g/kg Chitosan
Malt water (%)	4.5 ± 0.2 ^a^	4.3 ± 0.2 ^a^	4.4 ± 0.1 ^a^	4.4 ± 0.3 ^a^
Brittleness (%)	89.5 ± 0.5 ^a^	67.5 ± 0.4 ^b^	86.1 ± 0.5 ^a^	85.2 ± 0.4 ^a^
Kolbach Index (%)	42 ± 1 ^a^	38 ± 1 ^b^	42 ± 2 ^a^	43 ± 2 ^a^
α-amino nitrogen (%)	95 ± 3 ^b^	82 ± 2 ^c^	110 ± 4 ^a^	100 ± 4 ^b^
Extract (%)	77.5 ± 0.3 ^a^	54.3 ± 0.2 ^b^	77.8 ± 0.4 ^a^	76.8 ± 0.6 ^a^
Diastatic power (^o^WK)	198 ± 5 ^a^	165 ± 4 ^b^	204 ± 4 ^a^	198 ± 4 ^a^
Colority (EBC)	3.5 ± 0.0 ^b^	4.0 ± 0.0 ^a^	3.5 ± 0.0 ^b^	3.5 ± 0.0 ^b^
Turbidity (EBC)	0.4 ± 0.1 ^b^	1.5 ± 0.2 ^a^	0.5 ± 0.1 ^b^	0.6± 0.1 ^b^
β-glucan content (mg/L)	79 ± 3 ^b^	99 ± 3 ^a^	70 ± 2 ^c^	73 ± 2 ^c^
Viscosity (mPa·s)	1.4 ± 0.1 ^ab^	1.6 ± 0.1 ^a^	1.4 ± 0.1 ^ab^	1.4 ± 0.0 ^b^
Glucose content (g/L)	9.3 ± 0.2 ^a^	8.1 ± 0.1 ^b^	9.2 ± 0.2 ^a^	9.2 ± 0.1 ^a^
Maltose content (g/L)	45.6 ± 2.1 ^c^	35.4 ± 2.4 ^d^	54.8 ± 1.5 ^a^	50.2 ± 1.3 ^b^
Maltotriose content (g/L)	8.0 ± 0.2 ^a^	6.1 ± 0.1 ^b^	8.0 ± 0.3 ^a^	8.1 ± 0.5 ^a^
pH	6.0 ± 0.1 ^a^	5.8 ± 0.1 ^a^	5.9 ± 0.1 ^a^	6.0 ± 0.1 ^a^
Total acid (mL/100 mL)	1.4 ± 0.1 ^a^	1.5 ± 0.0 ^a^	1.5 ± 0.0 ^a^	1.4 ± 0.1 ^a^

Each number represents the mean ± standard deviation of three replicates. Mean values with different superscript letters in each row indicate a significant difference (*p* < 0.05).

## Data Availability

The data presented in this study are available on request from the corresponding author.

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
