# Peer review of "Inhibition of Chitosan with Different Molecular Weights on Barley-Borne Fusarium graminearum during Barley Malting Process for Improving Malt Quality"

_foods, 2022, doi:10.3390/foods11193058_

Round 1

Reviewer 1 Report

The manuscript by Luan et al. reports the evaluation of the inhibitory effects of chitosan on F. graminearum infection on barley malt. Moreover, they also optimized chitosan treatment efficacy on decontamination of malt in malting contaminated with F. graminearum and DON. The authors evaluate the effect of chitosan on barley germination and malt quality.

The manuscript is well written and presented. The experimental design is appropriate to the declared scope of the manuscript. The results are clearly reported. 

However, the Introduction and the Conclusion could be further improved:

1) In the Introduction section, the authors should describe the current methods used to inhibit pathogenic fungi or to reduce mycotoxins amount during food processing for improvement of malt quality. This could be useful to give more context, for instance: Are these methods less sustainable than the use of chitosan?

2) In the Conclusion section, more information about how the next steps would be to apply chitosan in the malting industry as an edible antifungal agent: Are Further investigations needed? Do the authors count to Scale-Up this experiment?

Author Response

Original manuscript ID: foods-1905314

Title: Inhibition of chitosan on barley-borne Fusarium graminearum during barley malting process for improving malt quality

Author(s): Jing Luan, Xu Wei, Zhefeng Li, Wenzhu Tang, Fan Yang, Zhimin Yu* and Xianzhen Li

Dear Ms. Mina Zhang and reviewers:

Thank you very much for the careful reading of our manuscript entitled "Inhibition of chitosan on barley-borne Fusarium graminearum during barley malting process for improving malt quality", and for the opportunity to submit a revised version. We sincerely appreciate the editor and reviewers for providing valuable suggestions and comments that have greatly helped us to improve the manuscript.

We have thoroughly considered all of the comments and have substantially revised our manuscript accordingly.

Attached, please find our point-by-point response to the reviewers’ comments. We would be most grateful if you could consider the thoroughly rewritten manuscript for publication foods.

Thank you and best regards.

Sincerely,

Zhimin Yu

School of Biological Engineering, Dalian Polytechnic University, Ganjingziqu, Dalian 116034, People’s Republic of China

Response to Reviewer 1 Comments:

Comments: The manuscript by Luan et al. reports the evaluation of the inhibitory effects of chitosan on F. graminearum infection on barley malt. Moreover, they also optimized chitosan treatment efficacy on decontamination of malt in malting contaminated with F. graminearum and DON. The authors evaluate the effect of chitosan on barley germination and malt quality. The manuscript is well written and presented. The experimental design is appropriate to the declared scope of the manuscript. The results are clearly reported. However, the Introduction and the Conclusion could be further improved:

Response: We appreciate the referee’s enthusiasm for our work. Thanks for these great suggestions. Below are our detailed point-to-point responses.

Point 1: In the Introduction section, the authors should describe the current methods used to inhibit pathogenic fungi or to reduce mycotoxins amount during food processing for improvement of malt quality. This could be useful to give more context, for instance: Are these methods less sustainable than the use of chitosan?

Response: “Inhibiting the infection and reproduction of Fusarium stain during malting is an effective measure to improve the quality of malt and the accumulation of mycotoxins in beer [12]. At present, the methods to inhibit the growth and reproduction of molds in barley mainly include spraying chemical fungicides [1], physical treatment [11] and adding antifungal microorganisms [2, 5]. However, excessive use of chemical fungicides can cause food safety issues and endanger workers' health. It is difficult to treat barley by radiation method on an industrial scale. Adding microorganisms in the malting process not only increases the production cost, but also may cause secondary pollution. Therefore, the development of environmentally friendly, efficient and inexpensive antifungal agents is a key problem to be solved urgently.” was added at the introduction section (p2, line 47-56), it is used to introduce the current methods used to inhibit malting mold, and introduce the shortcomings of these methods compared with the method of adding chitosan. 

Point 2: In the Conclusion section, more information about how the next steps would be to apply chitosan in the malting industry as an edible antifungal agent: Are Further investigations needed? Do the authors count to Scale-Up this experiment?

Response: “We found that chitosan can promote barley germination and improve malt quality, but its optimal technology conditions is currently unknown, and we will continue to study this in the future. At the same time, we will further expand the experimental scale to verify the application effect of chitosan, so as to better apply chitosan to the malting industry.” was added at the conclusion section (p.12, line 429-433).

Reviewer 2 Report

The authors presented the inhibition of chitosan on barley-borne Fusarium graminearum during barley malting process for improving malt quality. The idea of the article is good. Well designed. Appropriate analyses are measured and good discussions are provided. However, a few points to improve the current format of the article will be mentioned below:

The title could be modified to mention “various molecular weights”.

Minor editing of English language and style required. For example in line 10: There are many of -> There are many; etc.

It needs to be read carefully by an English speaker so that its grammatical problems are completely resolved.

Line 74: On what basis was the concentration of 1% of chitosan selected?

A comparison of mean data should be made between different days of the same concentration and between different concentrations on the same day. They must be displayed in uppercase and lowercase letters, separately.

Conclusion: what is the future of your findings? Conclusion is not insightful, what are suggestions?

Author Response

Original manuscript ID: foods-1905314

Title: Inhibition of chitosan on barley-borne Fusarium graminearum during barley malting process for improving malt quality

Author(s): Jing Luan, Xu Wei, Zhefeng Li, Wenzhu Tang, Fan Yang, Zhimin Yu* and Xianzhen Li

Dear Ms. Mina Zhang and reviewers:

Thank you very much for the careful reading of our manuscript entitled "Inhibition of chitosan on barley-borne Fusarium graminearum during barley malting process for improving malt quality", and for the opportunity to submit a revised version. We sincerely appreciate the editor and reviewers for providing valuable suggestions and comments that have greatly helped us to improve the manuscript.

We have thoroughly considered all of the comments and have substantially revised our manuscript accordingly.

Attached, please find our point-by-point response to the reviewers’ comments. We would be most grateful if you could consider the thoroughly rewritten manuscript for publication foods.

Thank you and best regards.

Sincerely,

Zhimin Yu

School of Biological Engineering, Dalian Polytechnic University, Ganjingziqu, Dalian 116034, People’s Republic of China

Response to Reviewer 2 Comments:

Comments: The authors presented the inhibition of chitosan on barley-borne Fusarium graminearum during barley malting process for improving malt quality. The idea of the article is good. Well designed. Appropriate analyses are measured and good discussions are provided. However, a few points to improve the current format of the article will be mentioned below:

Response: We appreciate the referee’s enthusiasm for our work. Thanks for these great suggestions. Below are our detailed point-to-point responses.

Point 1: The title could be modified to mention “various molecular weights”.

Response: The original title “Inhibition of chitosan on barley-borne Fusarium graminearum during barley malting process for improving malt quality” was changed as “Inhibition of chitosan with different molecular weights on barley-borne Fusarium graminearum during barley malting process for improving malt quality”.

Point 2: Minor editing of English language and style required. For example in line 10: There are many of -> There are many; etc.

Response: Thanks for the referee’s suggestion. We have carefully checked and revised the English language and style of this manuscript.

Point 3: It needs to be read carefully by an English speaker so that its grammatical problems are completely resolved.

Response: we have invited native English speaker to revise the language of this manuscript.

Point 4: Line 74: On what basis was the concentration of 1% of chitosan selected?

Response: I'm so sorry for the misunderstanding. This 1% is the concentration of the acetic acid solution used to dissolve the chitosan. Chitosan needs to be dissolved in 1%~2% acidic solution. In this paper, chitosan was firstly formulated into 40g/L original solution and then diluted for application. When the concentration of original chitosan solution exceeds 40g/L, it is difficult to dissolve chitosan and form gel, which is not conducive to application. “40 g chitosan was dissolved in 1 L 1%, (v/v) acetic acid solution...” was changed as “40 g chitosan was dissolved in 1 L of acetic acid solution (1%, v/v)...”.

Point 5: A comparison of mean data should be made between different days of the same concentration and between different concentrations on the same day. They must be displayed in uppercase and lowercase letters, separately.

Response: We have added the the significant difference of different conditions at the same time and different times for the same condition in Figure 1, Figure 2, Figure 3 and Figure 4, respectively.

Point 6: Conclusion: what is the future of your findings? Conclusion is not insightful, what are suggestions?

Response: “We found that chitosan can promote barley germination and improve malt quality, but its optimal technology conditions is currently unknown, and we will continue to study this in the future. At the same time, we will further expand the experimental scale to verify the application effect of chitosan, so as to better apply chitosan to the malting industry.” was added at the conclusion section (p.12, line 429-433).

Round 2

Reviewer 2 Report

The desired corrections have been made.